# Moisture Performance of Façade Elements Made of Thermally Modified Norway Spruce Wood

**Miha Humar ***[ID]**, Boštjan Lesar**[ID] **and Davor Kržišnik**[ID]

University of Ljubljana, Biotechnical Faculty, Department for Wood Science and Technology, Ljubljana SI1000, Slovenia; bostjan.lesar@bf.uni-lj.si (B.L.); davor.krzisnik@bf.uni-lj.si (D.K.)
* Correspondence: miha.humar@bf.uni-lj.si

**Abstract:** Wooden façades are gaining in importance. Thermally modified wood is becoming one of the preferred materials for claddings. In spite of the fact that façades made of thermally modified wood have been in use for more than two decades, reports about long-term monitoring have been sparse. The results of three-year monitoring of a façade made of thermally modified wood in Ljubljana are reported. Moisture content measurements of thermally modified façades were taken at 22 locations and compared to the moisture content of untreated Norway spruce wood. Temperature and relative humidity were recorded in parallel. The moisture content of the wood was compared to the average relative humidity before the measurements. The results confirm the lower moisture content of thermally modified wood in comparison to reference Norway spruce. The moisture content of the wooden façade could be best correlated with the average relative humidity and temperature 48 h before the wood moisture content measurement was taken.

**Keywords:** thermally modified wood; cladding; performance; sorption properties; water exclusion efficacy

## 1. Introduction

The construction sector consumes 40% of materials, 17% of freshwater and is responsible for 40%–50% of global greenhouse gas emissions [1]. One of the attempts to improve the environmental performance of this sector is to use more renewable materials. Wood is one of the most important renewable materials in construction applications due to its excellent mechanical properties and pleasing visual appearance. In addition to construction applications, wood in exterior applications also has to fulfil aesthetic requirements [2]. This is predominately important for façades. Low maintenance costs, broad accessibility and relatively simple production and mounting contribute to the increasing popularity of wooden façades. Furthermore, [3] showed that a wooden façade is favourable in terms of greenhouse gas emissions, compared to other materials such as brick, fibre cement and steel. Wooden façades can be made of untreated wood (predominately softwoods), wood treated with biocides or modified wood [4].

In recent decades, there has been a considerable increase in the use of thermally modified wood for cladding [5]. Modification of wood is defined as the persistent change of wood with the aim of increasing its inherent durability, as well as for enhancing its dimensional stability and other relevant properties [6]. The key benefit of thermally modified wood is to make abundant, non-durable wood species, such as pine, spruce, poplar or aspen, more durable and dimensionally stable. They thus perform more like durable wood species, at least in above-ground applications [4,7]. Thermal modification of wood can be considered as partial pyrolysis in a chamber with a low-oxygen concentration. The process results in a modified chemical composition of the wood. The first prominent degradation of hemicellulose starts at 140 °C and $\alpha$-cellulose at 150 °C [7]. Degradation and/or modification of lignin begins at higher

temperatures. One of the most crucial consequences of thermal modification is the reduction of readily available hydroxyl groups [8,9]. The equilibrium moisture content of the thermally modified wood is thus much lower than in non-modified wood when determined under the same climatic conditions [10]. The effect of the modification depends on the modification temperature and modification duration.

Fungi can degrade wood if the MC is above a specific limit. There are versatile data available in the literature. In the first set of the data, MC limits are stated to depend predominantly on the fungal species. Namely, Schmidt [11] reported that the minimum MC of wood varied between (30%) (*Fibriporia vaillantii* and *Gloeophyllum trabeum*) and 25% (*Coniophora puteana* and *Serpula lacrymans*). However, novel data indicate that that MC limits for fungal growth depend on the fungal species and wood species investigated as well. For example, limit MC for *C. puteana* growth on thermally modified Scots pine sapwood is 12.1%. Limit MC for fungal decay of thermally modified wood is 24.4% for thermally modified pine wood [12].

When wood is used in outdoor applications, it is essential to differentiate between the technical (functional) and the aesthetic service life. The technical service life of a façade is defined by the ability to withstand climatic impacts (wind-driven rain, other rain events, high relative humidity, potential condensation) and accounts for the ability of the wood to dry out [13]. The technical service life of a façade is greatly influenced by both the initial design and the material used. One of the most significant challenges is to assess the long-term performance of the various materials in less exposed applications, such as cladding. The decay under use class 3.1 conditions [14] is considerably slower than in frequently used above ground tests (e.g., lap-joint, double-layer tests) and in-ground tests [15]. Laboratory tests are a faster alternative to field testing, but which are often too severe and not all parameters influencing the degradation can be treated adequately. It is therefore of great importance to develop a methodology for the fastest assessment of the performance of wood and wood-based materials in less exposed applications. Published studies have indicated that moisture performance field tests may serve as a time-saving alternative to long-term decay tests in the field [4,16,17]. This approach is supported by a model approach that defines material resistance based on the combined effect of wetting ability and durability [18]. The idea of this study was to assess the performance of a façade made of thermally modified Norway spruce wood with three-year moisture monitoring. Although modified wood has been on the market for two decades, comprehensive moisture monitoring studies of the modified wood in use are rare. To the best of our knowledge, this is the first study in which the moisture content of wood has been monitored for several years at multiple locations within a representative façade.

## 2. Material and Methods

Figure 1 shows the wooden Annex to the existing building of the Department of Wood Science and Technology in Ljubljana, Slovenia (46°02′55.7″N 14°28′47.3″E, elevation above mean sea level 293 m) finished in December 2015. In Ljubljana, the average temperature is 10.4 °C, annual precipitation 1290 mm and Scheffer Climate Index 55.3. The Annex has a rectangular shape with dimensions 40 m × 13.5 m. As the most important educational and research institution in the field of wood in Slovenia, this also serves learning and demonstration purposes. The construction is made of cross-laminated timber (CLT) boards, while the façade is made of thermally modified spruce wood (Thermowood, Thermo D). The sloping non-structural outer columns are also made of thermally modified spruce, giving the sense of a forest through the extensive glazing on the longitudinal southern and northern façades.

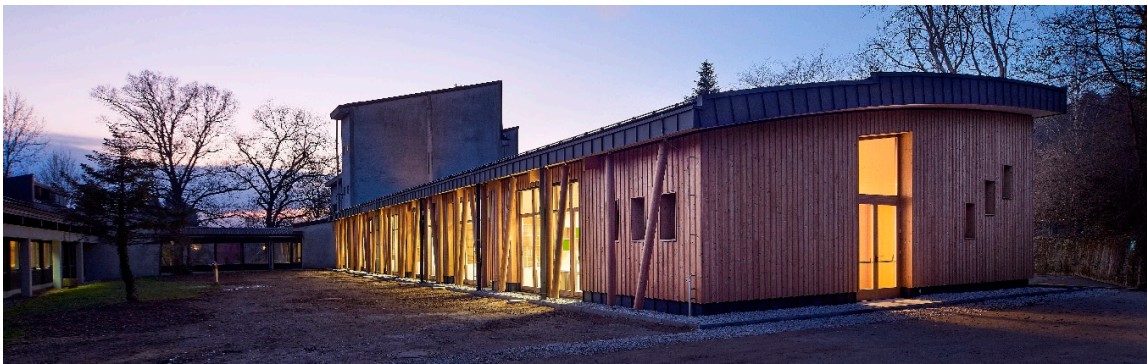

**Figure 1.** Annex to the existing building of the Department of Wood Science and Technology in Ljubljana (photo: Damjan Švarc).

To determine the relationship between relative humidity (RH) and moisture content (MC) of thermally modified wood, dynamic water vapour sorption (DVS) was measured. Samples for DVS analysis were isolated from the façade, to have a representative sample. Wood was milled on an IKA mill (A 11 basic Analytical mill, Staufen, Germany). Before the experiment, the wood chips were conditioned for 48 h at 1 ± 1% RH and 20 ± 0.2 °C. Sorption analysis of the thermally wood samples was performed using a Dynamic Vapour Sorption (DVS Intrinsic, Surface Measurement Systems Ltd., London, UK). Pre-conditioned wood chips (approximately 40 mg) were placed on the sample holder. The holder was suspended in a microbalance within a sealed thermostatically controlled chamber. In the chamber, a constant flow of dry compressed air was passed over the sample (200 cm$^3$/s) at constant temperature (25 ± 0.2 °C). The method for DVS was set to 20 steps between 0% and 95% RH for sorption and desorption. DVS maintained a target RH until the weight change of the sample was less than 0.002%/min for 10 min. Relevant parameters were recorded every 60 s throughout the isotherm run. Sorption and desorption isotherms were produced by plotting equilibrium moisture content (EMC) change against relative humidity (RH).

The relation between wood MC and climate was also determined under real conditions. Monitoring of the material-climate conditions began on the 13th of July 2016. Two types of continuous monitoring were performed on the building. (1) Scanntronik's (Mugrauer GmbH) temperature sensors were positioned close to the wood moisture measurement sensors to obtain exact values for the conversion of electrical conductivity into wood moisture content (MC) [19]. Temperature (T) was monitored on the surface, not to interfere with MC measurements. (2) MC was determined through measurements of electrical resistance (Table 1). Insulated electrodes (stainless steel screws) were applied 5 mm below the surface of the wooden planks at various positions and linked to electrical resistance measuring devices Gigamodule (Scanntronik Mugrauer GmbH, Germany) (Figure 2) [17]. All measurements were performed on the south-facing façade, as this façade is more exposed to weathering. The majority of the wind-driven rain is coming from the south-east. This part of the building is more exposed, thus it was chosen for monitoring. The equipment used enables wood MC measurements in the range between 6% and 60%. MC was logged twice per day, at noon and midnight. To transfer electrical resistance measured inside the wood, wood species-specific resistance characteristics were developed. For determination of the characteristics for electrical resistance-based MC measurements, five replicates of wood samples made of Norway spruce and Thermally modified Scots pine (150 × 50 × 25 mm$^3$) were prepared. In each respective sample, two stainless-steel screws were positioned the same way as described for MC monitoring. In the beginning, wood samples were oven-dried (24 h; 103 ± 2 °C) and afterwards, their mass was determined. Samples were placed in a climatic chamber (Kambič, Slovenia) with variable parameters (temperature and relative humidity (RH)) to condition samples to constant weight for three weeks. After conditioning, their mass and electrical resistance were measured. Measurements were performed the same way as MC monitoring of the façade. The authors are aware that there is a limited moisture gradient formed within the

samples, but long conditioning times limit the gradients to a level suitable for preparation of calibration curves. Samples were weighed eight times (at 25%, 40%, 60%, 80%, 90%, 100%, after impregnation with distilled water and after drying at 100% RH) and at four temperatures (30 °C, 25 °C, 20 °C, and 15 °C) except for the final drying at 100% RH. Calibration curves were prepared based on the published methodology [19–21]. Use classes of wood were determined. Use Class is a term taken from EN 335 [14]. This European Standard defines five use classes that represent different service situations to which wood and wood-based products can be exposed:

- Use Class 1—Situations in which the wood or wood-based product is inside a construction, not exposed to the weather and wetting.
- Use Class 2—Situations in which the wood or wood-based product is undercover and not exposed to the weather (mainly rain and driven rain) but were occasional, but not persistent, wetting can occur.
- Use Class 3 Situations in which the wood or wood-based product is above ground and exposed to the weather (mainly rain). UC 3.1 is less exposed than 3.2.
- Use Class 4—A situation in which the wood or wood-based product is in direct contact with the ground and/or freshwater.
- Use Class 5—A situation in which the wood or wood-based product is permanently or regularly submerged in saltwater [14].

**Table 1.** Descriptions of the moisture content monitoring locations on the façade of the Annex of the Wood Science and Technology building in Ljubljana. Use classes are defined according to EN 335 [14].

| Abbr. | Material | Description | Use Class | No. of Meas. |
|---|---|---|---|---|
| PaTM 1a | Spruce TM | Plank 3.25 m above ground | UC2 | 2373 |
| PaTM 1b | Spruce TM | Plank 3.25 m above ground | UC2 | 2146 |
| PaTM 2a | Spruce TM | Plank 2.2 m above ground | UC2 | 2146 |
| PaTM 2b | Spruce TM | Plank 2.2 m above ground | UC2 | 2146 |
| PaTM 3a | Spruce TM | Plank 1.15 m above ground | UC2 | 2146 |
| PaTM 3b | Spruce TM | Plank 1.15 m above ground | UC2 | 2146 |
| PaTM 4a | Spruce TM | Plank 0.4 m above ground | UC2 | 2146 |
| PaTM 4b | Spruce TM | Plank 0.4 m above ground | UC2 | 2146 |
| PaTM 5 | Spruce TM | Pillar 3.25 m above ground front side | UC3.1 | 2232 |
| PaTM 6 | Spruce TM | Pillar 3.25 m above ground back side | UC3.1 | 2232 |
| PaTM 7 | Spruce TM | Pillar 2.2 m above ground front side | UC3.1 | 2232 |
| PaTM 8 | Spruce TM | Pillar 2.2 m above ground back side | UC3.1 | 2232 |
| PaTM 9 | Spruce TM | Pillar 1.15 m above ground front side | UC3.1 | 2232 |
| PaTM 10 | Spruce TM | Pillar 1.15 m above ground back side | UC3.1 | 2232 |
| PaTM 11 | Spruce TM | Pillar 0.2 m above ground front side | UC3.1 | 2232 |
| PaTM 12 | Spruce TM | Pillar 0.2 m above ground back side | UC3.1 | 2232 |
| PaTM 13a | Spruce TM | Window sill top | UC2 | 2186 |
| PaTM 13b | Spruce TM | Window sill side | UC2 | 2186 |
| PaTM 13c | Spruce TM | Window sill side | UC2 | 2186 |
| PaTM 13d | Spruce TM | Window sill | UC3.1 | 2186 |
| PaTM 13e | Spruce TM | Window sill | UC3.1 | 2186 |
| PaTM 13f | Spruce TM | Window sill | UC2 | 2186 |
| Pa 1a | Spruce | Roof beam, surface | UC2 | 2086 |
| Pa 1b | Spruce | Roof beam, surface | UC2 | 2086 |
| Pa 2a | Spruce | Roof beam, 2 cm deep | UC2 | 2086 |
| Pa 2b | Spruce | Roof beam, 5 cm deep | UC2 | 2086 |
| Pa 3 | Spruce | Celling surface | UC2 | 2086 |
| Pa 4a | Spruce | Celling 2 cm deep | UC2 | 2086 |
| Pa 4b | Spruce | Celling 2 cm deep | UC2 | 2186 |
| Pa 5 | Spruce | Celling 4 cm deep | UC2 | 2086 |
| Pa 6a | Spruce | Celling 5 cm deep | UC2 | 2086 |
| Pa 6b | Spruce | Celling 5 cm deep | UC2 | 2185 |

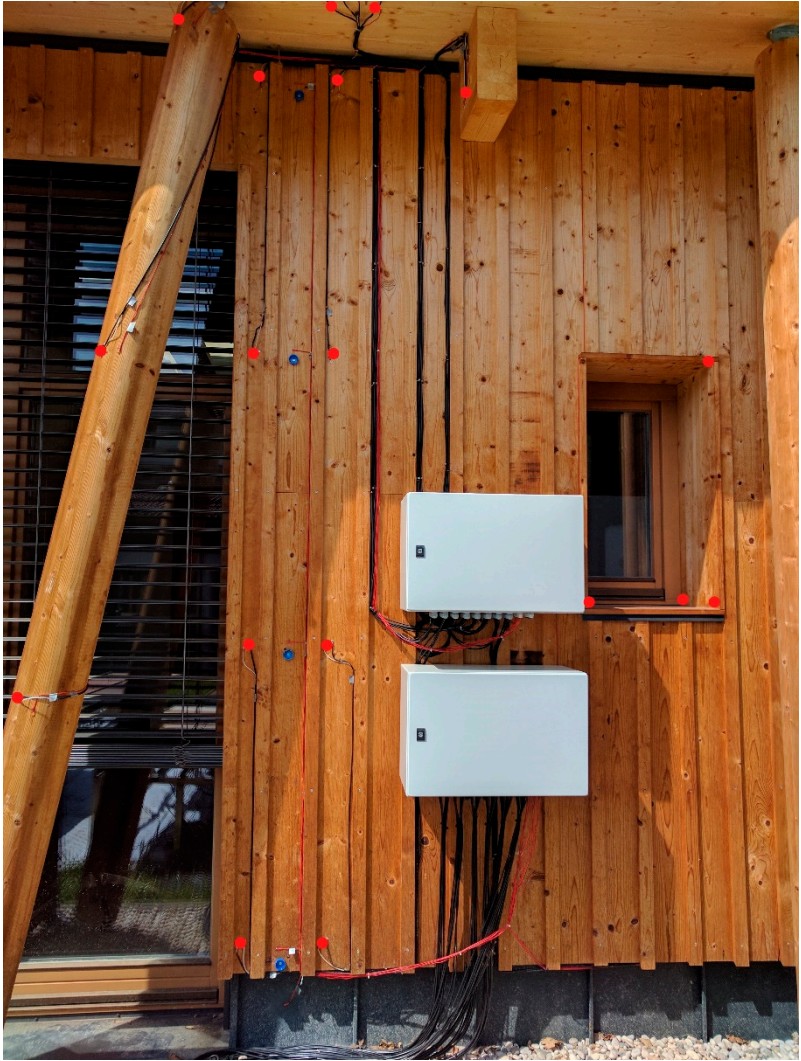

**Figure 2.** Position of wood moisture and temperature sensors on the façade of the Annex. The red dots indicate locations of the measurements.

Climatic conditions were determined at a nearby (distance 20 m) weather station (Vantage Pro, Davis Instruments, USA). Temperature (T) and relative humidity (RH) were logged every 15 minutes in the monitoring period between 13th of July 2016 and 15th of July 2019. Based on these data, average T and RH were calculated for periods between MC measurements (12 h period). Wood moisture content is predominately influenced by RH; a correlation between RH and wood MC was, therefore, determined, namely: average RH (average RH 12 h before MC measurements), two days' average RH (average RH 48 h before MC measurements) and three days' average RH (average RH 72 h before MC measurements). These periods were selected based on a preliminary analysis. In addition to RH, wood MC is also influenced by temperature. An additional indicator that corresponds to the theoretical MC of untreated wood was therefore calculated (f(RH; T)) (Eq. 1) [22]. The average RH 48 h and average temperature 48 h before MC was calculated for this indicator. It provides an overestimated MC of thermally modified wood; we therefore consider it as an indicator and not as actual wood moisture content.

$$f(RH;T) = \left(-0.000612T\left(1 - \frac{T}{647.1}\right)^{2.43}\ln(1 - RH)\right)^{0.0577T^{0.430}} \tag{1}$$

T—temperature in °K; RH—relative humidity

Pearson correlation coefficients (r) were calculated to determine the level of the strength of the linear relationship between wood MC and calculated indicators (Average RH, 48 h average RH, 72 h average RH and f(RH;T)) using GraphPad Prism, version 8.3.1. (GraphPad Software, San Diego, CA, US). The multiple-variable analysis procedure was used to calculate the correlation coefficients to measure the strength of the linear relationship between all the variables. The significance of the correlations was calculated as well.

## 3. Results and Discussion

Temperature, together with moisture content, is one of the critical factors affecting wood's service life. Too high or too low temperatures can affect fungal vitality and thus degradation of wood. The minimum temperature for fungal growth and degradation is usually around 3 °C. Below freezing point, there is no liquid water available for fungal metabolism. The optimal temperatures for development of decay fungi are between 20 and 30 °C. The maximal temperature for mycelial growth and wood degradation is generally between 40 and 50 °C. At higher temperatures, proteins (enzymes) start denaturing [11]. However, when interpreting the temperature data, it should be noted that temperatures at micro-locations can differ significantly, for instance, indicated by Gobakken et al. [23], who monitored wooden hunting huts in Svalbard, Norway.

As can be seen from Figure 3, air temperature varied between −14.8 °C (Januar 2017) and 36.6 °C (August 2017). Average temperatures (11.2 °C) were comparable in the three years. In the monitoring period, the air temperature was suitable for decay for 77.0% of the time. In general, the air temperature was below 3 °C for 21.8% of the time and above 30 °C for 1.3% of the time. However, air temperature cannot be directly transferred to the material temperature. As can be seen from Figure 4, temperatures on the façade were considerably higher than air temperatures. This phenomenon has already been reported [23]. The highest measured temperature on the façade was 58.6 °C, on the lowest part of the façade. This part is most exposed to solar radiation. However, it should be noted that this was the surface temperature only; the actual material temperatures in the central part of the planks might be slightly different.

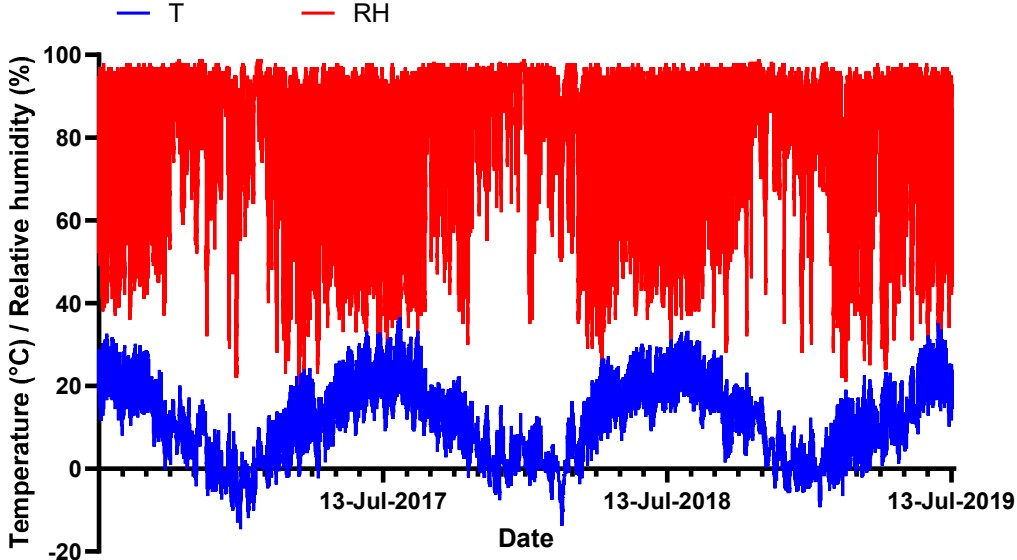

**Figure 3.** Temperature (T) and relative humidity (RH) in the nearby weather station between 13th of July 2016 and 15th of July 2019. Each curve represents 105,000 measurements.

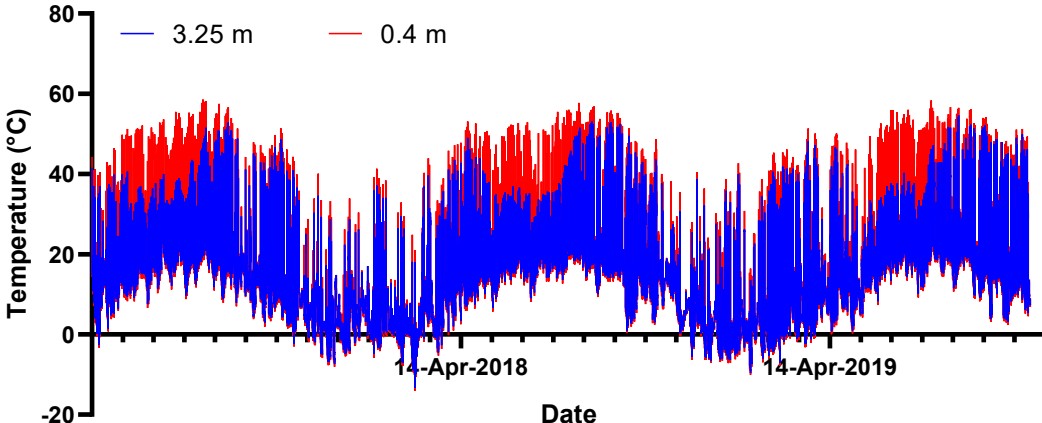

**Figure 4.** Influence of distance from the ground on the temperature dynamics on the façade in the period between 13th of July 2016 and 15th of July 2019. Each curve represents 22,300 measurements.

The temperatures at actual micro-locations varied. Differences on the object originate predominately from exposure to solar radiation. The percentage of days in the period between 13th of July 2016 and 15th of July 2019 (1098 days) with a façade temperature below 3 °C was 15.1%, which is a much lower percentage than presumed from the air temperature (21.8%). The higher temperature of the façade results from solar radiation and temperature losses through the walls (in winter). From Figure 4, the influence of the distance of the measurement location from the ground can be seen. Measurements were performed on the same vertical plank, under the roof and close to the ground. The upper part was shaded at noon, while the lower part of the plank was exposed to sunlight for an extended period of time during the day. There is not, therefore, much difference between the temperature dynamic during the coldest periods. On the other hand, differences between air temperature and the temperature of the façade became prominent during the summer.

Relative air humidity is another important factor that influences the performance of wood. If the RH is high, wood absorbs water vapour from the air. The importance of air humidity increases if air temperature falls below or close to the dew point. On wood located in conditions with high RH, the first staining fungi will occur at an RH above 80%, while higher RH, above 90%, is required for degradation [24]. However, for severe degradation, water traps and/or a condensing environment are necessary. RH in front of the Annex varied between 16.0% (April 2018) and 99.0% (Figure 3). There were fairly marked differences determined daily.

The relation between wood MC and RH can be seen from Figure 5. Wood MC increases with increasing RH. If the MC of Norway spruce is compared to the MC of thermally modified Norway spruce, it can clearly be seen that the MC of thermally modified wood at a given RH is considerably lower. For example, the MC of Norway spruce at 95% RH was 23.79%. On the other hand, the MC of thermally modified wood at the given RH was only 11.86% (Figure 5). This can be ascribed to a reduced number of hydroxyl groups with thermally modified wood [8,9]. However, it should be considered that DVS analysis could result in errors in the prediction of EMC [25,26]. However, the differences in EMC between TM wood and untreated wood are higher than respective error.

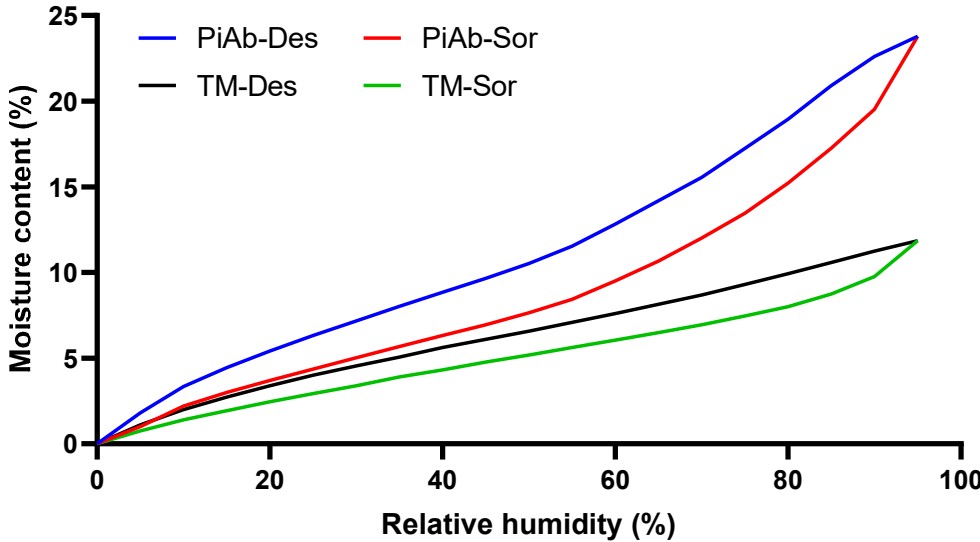

**Figure 5.** Relationship between relative humidity and wood moisture content, as determined with DVS for Norway spruce (PiAb) and thermally modified Norway spruce (TM) in sorption (Sor) and desorption (Des) cycles.

Aggregated wood MC data are presented in Table 2 and Figure 6. As expected, the wood MC reflected the micro-climate in the environment. Moreover, the MC of thermally modified wood was significantly lower than the MC of non-modified Norway spruce (Table 2). The average MC of spruce was 13.8%, while MC of thermally modified wood was considerably lower (8.2%), even though the thermally modified wood was more exposed to wind-driven rain than the spruce. This is a consequence of the reduction of readily available hydroxyl groups during thermal modification [8] and it is in line with DVS observations (Figure 5). The MC of Norway spruce wood at 60% RH in desorption was 12.83%, while that of thermally modified wood was 7.61%. The ratio between the MC of TM wood and Norway spruce wood at 60% RH (1.685) is comparable to the ratio between the average MC on the façade (1.682) between spruce and TM spruce on the respective façade. This clearly indicates that the sorption properties of materials and RH have a prevalent influence on the MC of wood.

**Table 2.** Wood moisture content data in the locations of the building at which monitoring was performed between 13th of July 2016 and 15th of July 2019. Abbreviations are the same as in Table 1.

| Abbr. | Moisture Content (%) | | | No. of Meas. Above Threshold MC | |
| --- | --- | --- | --- | --- | --- |
| | Average | Median | Max | 20% | 25% |
| PaTM 1a | 6.8 | 6.6 | 11.9 | 0 | 0 |
| PaTM 1b | 7.6 | 7.4 | 12.0 | 0 | 0 |
| PaTM 2a | 6.8 | 6.4 | 16.9 | 0 | 0 |
| PaTM 2b | 7.7 | 7.2 | 20.1 | 1 | 0 |
| PaTM 3a | 6.5 | 6.4 | 19.5 | 0 | 0 |
| PaTM 3b | 7.9 | 7.3 | 24.6 | 3 | 0 |
| PaTM 4a | 7.2 | 6.8 | 27.3 | 2 | 1 |
| PaTM 4b | 8.0 | 7.3 | 24.7 | 3 | 0 |
| PaTM 5 | 8.4 | 8.3 | 14.5 | 0 | 0 |
| PaTM 6 | 8.1 | 8.0 | 14.5 | 0 | 0 |
| PaTM 7 | 7.0 | 6.6 | 30.8 | 5 | 1 |
| PaTM 8 | 8.1 | 8.2 | 15.8 | 0 | 0 |
| PaTM 9 | 8.4 | 8.0 | 29.7 | 13 | 7 |
| PaTM 10 | 9.0 | 8.7 | 18.0 | 0 | 0 |
| PaTM 11 | 8.5 | 7.2 | 34.9 | 65 | 35 |
| PaTM 12 | 7.2 | 6.8 | 21.5 | 1 | 0 |
| PaTM 13a | 9.3 | 8.5 | 30.4 | 18 | 1 |
| PaTM 13b | 8.1 | 7.8 | 15.3 | 0 | 0 |
| PaTM 13c | 7.7 | 7.5 | 18.2 | 0 | 0 |
| PaTM 13d | 11.6 | 9.3 | 47.5 | 230 | 67 |
| PaTM 13e | 13.1 | 11.6 | 36.0 | 315 | 117 |
| PaTM 13f | 8.2 | 7.7 | 18.9 | 0 | 0 |
| Pa 1a | 16.0 | 16.2 | 23.3 | 123 | 0 |
| Pa 1b | 13.1 | 13.8 | 20.7 | 1 | 0 |
| Pa 2a | 14.5 | 14.6 | 17.6 | 0 | 0 |
| Pa 2b | 14.4 | 14.5 | 17.7 | 0 | 0 |
| Pa 3 | 14.2 | 14.4 | 17.6 | 0 | 0 |
| Pa 4a | 14.1 | 14.4 | 17.4 | 0 | 0 |
| Pa 4b | 13.8 | 13.9 | 17.6 | 0 | 0 |
| Pa 5 | 12.8 | 13.2 | 16.1 | 0 | 0 |
| Pa 6a | 12.8 | 13.2 | 17.3 | 0 | 0 |
| Pa 6b | 11.9 | 12.2 | 15.5 | 0 | 0 |

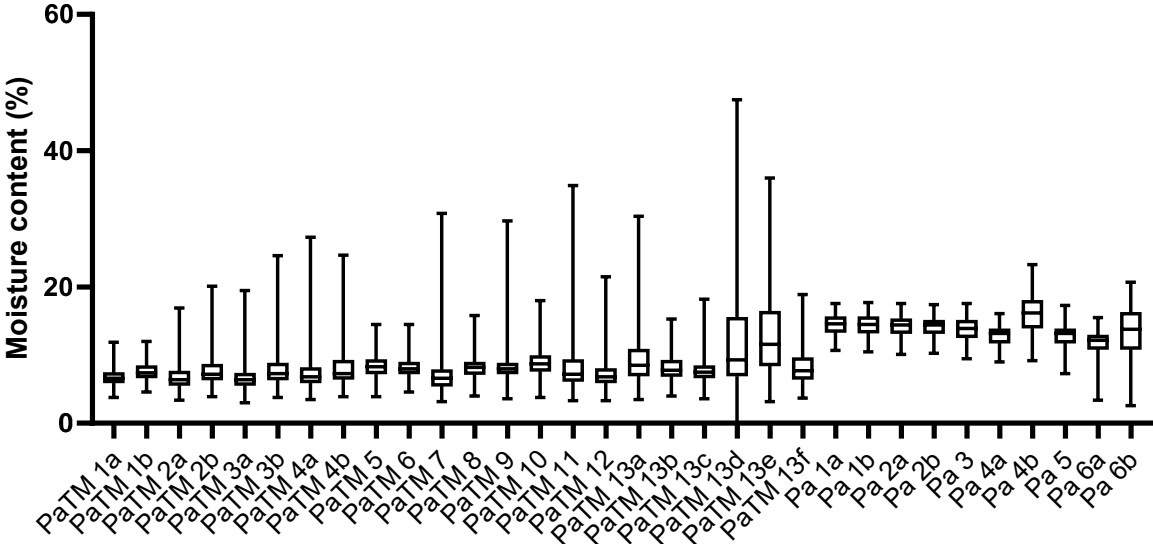

**Figure 6.** Wood moisture content at different monitoring positions on the façade of the Annex of the Wood Sci. and Tech. building. The box displays the mean values (25th and 75th percentile), while the whiskers stretch to the highest and lowest value. The abbreviations are the same as in Table 1.

The MC of thermally modified wood was fairly uniform. The lowest average MC was 6.8% (PaTM 1a and PaTM 2a). Those values were determined on the same plank, 3.25 m and 2.2 m above

ground (Table 1). These parts were protected from wind-driven rain with a roof overhang. On the other hand, the highest average MC was determined on horizontal planks of a window sill (PaTM 13e; 13.1%). The primary reason for the high MC is horizontally positioned planks, with a water trap. The water thus stays on the surface of the plank, which results in the highest average and median values. In addition to the window sill, a higher MC was also determined on the pillars. They were positioned at the edge of the roof (Figure 1). However, it should be noted that all MC measurements on thermally modified wood were performed 5 mm below the wood surface. Surface wood MC might have been even higher.

The influence of micro-location on wood MC can be seen from Figure 7. Moisture content was measured on the same plank on different heights. As expected, the lowest MC was determined at the top location, 3.25 m above ground. MC determined at the other places seldom exceeded 20%. MC increased above the average value depending on the direction of wind-driven rain. Due to the roof-overhang and sheltering of the columns, the façade was not frequently exposed to rain events.

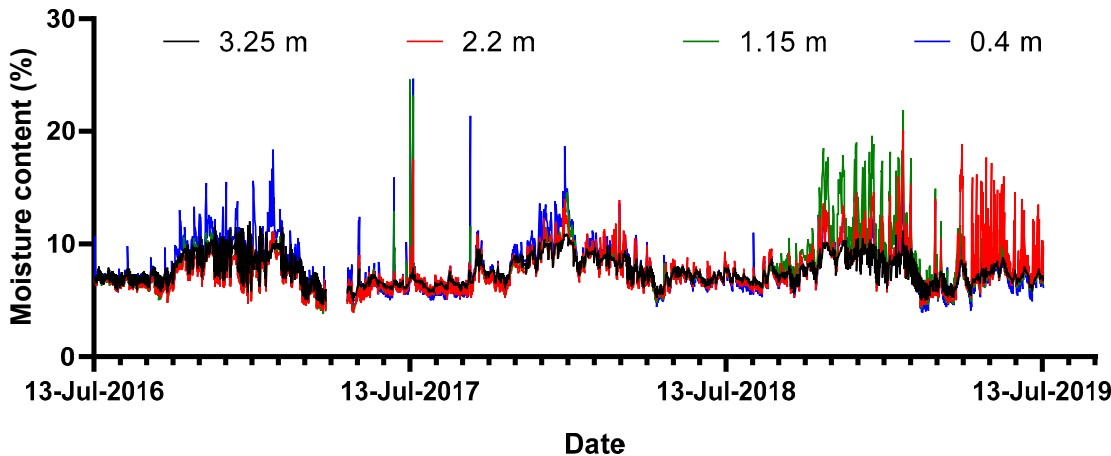

**Figure 7.** Influence of the distance from the ground on the wood moisture content of the façade on the monitored Annex of the Wood Science and Technology building in Ljubljana in the period between 13th of July 2016 and 15th of July 2019.

In addition to average, median and extreme data, the percentage of wet days, i.e., days when wood MC exceeded a particular threshold, are reported in Table 2. Different thresholds were taken into account. In general, the 25% MC threshold is considered to be the minimum MC required for fungal decay of untreated wood. This generally represents a conservative fibre saturation value. However, lower values are possible if the fungi can transport water from a neighbouring moisture source to the wood [12]. The threshold values for thermally modified wood correspond to the hygroscopic properties and corresponding fibre saturation. Fibre saturation of thermally modified wood is lower than reported for untreated wood, as seen in Figure 7 [4,7]; the MC threshold is thus also lower [12]. A fairy high number of measurements exceeding the limit of 20% for modified wood were determined on the front side of the wooden pillars, which were the most exposed to rain events (PaTm 7, PaTm 9, and PaTm 11). Even higher values were determined with horizontally oriented elements of the window sill (PaTm 13d and PaTm 13e). Water was not able to run off from the sill; it therefore accumulated in the wood, which resulted in increased wood MC levels. However, although these values were higher than determined on the rest of the façade, it should be noted that thermally modified wood has much better inherent durability; it can thus withstand a higher number of "wet-days" compared to less durable Norway spruce or Scots pine sapwood [4,18].

One of the objectives of this study was to address the influence of climatic conditions on wood MC. Since the majority of the façade is designed in a way that it is not exposed to rain, except for wind-driven rain, we decided to focus on the correlation between RH and wood MC. This relation

is fairly straightforward in the laboratory, with stable, controlled conditions. As can be seen from Figure 3, under real conditions, relative humidity varied significantly through the day. The average difference between the daily highest and the daily lowest RH was 26%. The highest daily difference in RH of 69% was recorded in January 2017 and February 2018. To consider daily variations fully, we compared the average RH in the previous 12 h, 48 h and 72 h with wood MC. Pearson's correlations are calculated in Table 3. MC of non-modified Norway spruce correlated best with average RH during the previous 12 h. On the other hand, MC of thermally modified wood correlated best with the 48 h average RH before the specific moisture measurements. This suggests that thermally modified wood takes a longer time to react to changes in RH [9].

**Table 3.** Coefficient of correlation between wood MC and selected indicators: Average RH (average RH 12 h before MC measurements), 48 h average RH (average RH 48 h before MC measurements), 72 h average RH (average RH 72 h before MC measurements), f(RH; T) (value calculated according to Equation (1), considering average RH 48 h and average temperature 48 h before MC measurements). All corelations are statisticaly significant (*p* < 0.0001).

| Abbr. | Correlation Factors | | | |
| --- | --- | --- | --- | --- |
| | Average RH | 48 h Average RH | 72 h Average RH | f(RH;T) |
| PaTM 1a | 0.56 | 0.71 | 0.72 | 0.76 |
| PaTM 1b | 0.57 | 0.64 | 0.70 | 0.70 |
| PaTM 2a | 0.44 | 0.65 | 0.63 | 0.73 |
| PaTM 2b | 0.48 | 0.61 | 0.59 | 0.65 |
| PaTM 3a | 0.42 | 0.64 | 0.63 | 0.70 |
| PaTM 3b | 0.44 | 0.59 | 0.58 | 0.65 |
| PaTM 4a | 0.43 | 0.63 | 0.63 | 0.72 |
| PaTM 4b | 0.45 | 0.65 | 0.66 | 0.73 |
| PaTM 5 | 0.51 | 0.65 | 0.64 | 0.69 |
| PaTM 6 | 0.59 | 0.71 | 0.71 | 0.76 |
| PaTM 7 | 0.24 | 0.56 | 0.49 | 0.59 |
| PaTM 8 | 0.56 | 0.69 | 0.66 | 0.71 |
| PaTM 9 | 0.37 | 0.50 | 0.47 | 0.52 |
| PaTM 10 | 0.50 | 0.68 | 0.64 | 0.72 |
| PaTM 11 | 0.36 | 0.55 | 0.49 | 0.59 |
| PaTM 12 | 0.41 | 0.67 | 0.63 | 0.73 |
| PaTM 13a | 0.49 | 0.68 | 0.64 | 0.74 |
| PaTM 13b | 0.54 | 0.80 | 0.79 | 0.84 |
| PaTM 13c | 0.51 | 0.58 | 0.62 | 0.61 |
| PaTM 13d | 0.48 | 0.62 | 0.60 | 0.66 |
| PaTM 13e | 0.56 | 0.71 | 0.68 | 0.77 |
| PaTM 13f | 0.55 | 0.79 | 0.77 | 0.84 |
| Pa 1a | 0.73 | 0.60 | 0.68 | 0.71 |
| Pa 1b | 0.73 | 0.63 | 0.72 | 0.30 |
| Pa 2a | 0.61 | 0.21 | 0.29 | 0.32 |
| Pa 2b | 0.64 | 0.24 | 0.31 | 0.70 |
| Pa 3 | 0.71 | 0.61 | 0.70 | 0.46 |
| Pa 4a | 0.70 | 0.38 | 0.48 | 0.62 |
| Pa 4b | 0.53 | 0.58 | 0.58 | 0.59 |
| Pa 5 | 0.75 | 0.53 | 0.59 | 0.23 |
| Pa 6a | 0.67 | 0.20 | 0.28 | 0.61 |
| Pa 6b | 0.46 | 0.56 | 0.59 | 0.61 |

The best correlation between the average RH and thermally modified wood was determined with elements that were sheltered and not exposed to wind-driven rain, such as the upper parts of the façade (PaTm 1a) and the sheltered part of a window sill (PaTm 13b, and 13f), where correlations of 0.71, 0.79 and 0.80 were determined, respectively (Table 3). Since the wood MC is a factor of RH and T, the indicator calculated based on Equation (1) [27] was also considered. This indicator correlates on average with the MC of thermally modified wood better than RH alone. The highest correlation of 0.82 was determined at the sheltered part of the window sill (PaTm 13b) (Figure 8). This result indicates that this indicator can be used for evaluation of MC of thermally modified wood in outdoor, sheltered conditions (UC 2; EN 335 [14]). It should be noted that the correlation between non-modified

spruce wood and the respective indicators is much more challenging to perform in the present study, since the MC with the spruce wood was determined at various depths. In addition, there could even be an adhesive line between the layers in CLT, which also influences vapour diffusion.

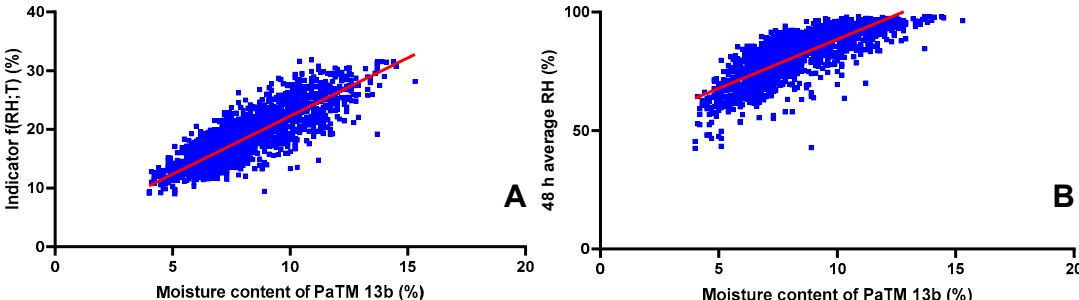

**Figure 8.** Correlation between MC of thermally modified wood (window sill, PaTM 13b) and selected indicators: (**A**) Value calculated according to Equation (1), considering average RH 48 h and average temperature 48 h before MC measurements (f(RH; T)) (R = 0.84; *p* < 0.0001); (**B**) Two-day average RH (48 h average RH) (*R* = 0.80; *p* < 0.0001). The trend line is red.

All correlations presented in Table 3 are statistically significant (*p* < 0.0001). Statistical significance indicates that the idea presented is relevant and reflect the high number of the measurements performed. We are planning to continue with the monitoring of the respective building to additionally validate the concept and asses long term performance of thermally modified wood.

## 4. Conclusions

The moisture content of thermally modified Norway spruce wood on the cladding was considerably lower than the moisture content of untreated reference Norway spruce. This is in line with the performed dynamic vapour studies. The high moisture content of the wooden façade elements is associated with increased exposure to wind-driven rain or water traps. The moisture content of thermally modified wood correlates with an indicator calculated from average relative humidity and temperature 48 h before the wood moisture content measurement. The three-year monitoring proved that thermally modified wood is an excellent choice for a wooden façade.

**Author Contributions:** Conceptualization, M.H.; methodology, M.H. and D.K..; validation, M.H., D.K. and B.L.; formal analysis, M.H. and D.K.; investigation, M.H.; resources, M.H.; data curation, M.H..; writing—original draft preparation, M.H.; writing—review and editing, B.L and D.K.; visualization, M.H..; supervision, M.H.; project administration, M.H. and B.L.; funding acquisition, M.H. All authors have read and agreed to the published version of the manuscript.

**Funding:** The authors acknowledge the support of the Slovenian Research Agency (ARRS) within the framework of research projects L4-7547, research program P4-0015, and the infrastructural centre (IC LES PST 0481-09). Part of the presented research was also supported by the project: Sustainable and innovative construction of smart buildings - TIGR4smart (C3330-16-529003) and Wood and wood products over a lifetime (WOOLF).

**Conflicts of Interest:** The authors declare no conflict of interest.

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
