# Peer review of "Moisture Performance of Façade Elements Made of Thermally Modified Norway Spruce Wood"

_forests, doi:10.3390/f11030348_

Round 1

Reviewer 1 Report

The authors have addressed all of my previous comments. I appreciate their efforts in addressing my comments. The paper is scientifically sound.

Author Response

Dear reviewer, thank you for your positive response. 

Sincerely, 

Miha Humar 

Reviewer 2 Report

The article is very interesting and important for wood industry and economy. I have read the manuscript with pleasure and I hope that you will continue your studies in this field. In order to improve the quality of your paper I recommend you a few minor revisions. line 40: thermally modifying wood, not thermally modified wood as in the whole article? lines 40-41: "non-durable wood species" (and many times in the article) isn't too much this "no"? I would say less durable. line 53: Please, add an "i", I think that here should be "Determination of respective limit is rather..." line 83 and many times in the article: I wouldn't write "Annex", just "annex". line 119: Please, explain why did you choose the south-facing façade. line 152: Equation 1 is a function of T and h in order to provide an estimate for MC. Why do you write f(MC; T) here and many times in the article? I don't understand. Why did you use h as symbol for relative humidity and not RH? I would use f(RH, T). line 155: Why did you use T in Kelvin? line:156: Please, add the unit for relative humidity (%). line 166: Please, change the order of words, maximum temperature, not "temperature maximum". lines 189, 190, 306: Please, use the same font and the same size. line 206: Figure 5, not Figure 5 5. line 233: I would add "it" after "and" to be "and it is in line". lines 235-237: I think that this sentence requires an addition at the end. "ratio between the average MC" and?? Figure 8: I think that a curve is better to be used in order to describe the relationship in chart B.

Author Response

see attachment... 

This manuscript is a resubmission of an earlier submission. The following is a list of the peer review reports and author responses from that submission.

Round 1

Reviewer 1 Report

This paper presents moisture monitoring data on thermally modified facades. This is an important area of research with little data. Overall the paper is sound. I have some concerns with the experimental methods that I am hoping the authors can address.

My biggest concern is the calibration for the electric moisture meters. Since gravimetric measurements are not presented in the paper, it is essential that the measurements are appropriately calibrated for this material. The paper references work by Otten that did look at thermally modified spruce. However, it is not clear if the calibration function of Otten was used or if the authors took their own calibration measurements. More detail is needed on this section. It may also be helpful if the relationships between the resistance and moisture content were reported so that others could in theory, validate this work.

The 0.002% min criteria has been shown to give extremely large errors in the prediction of EMC  (as high as 1.8% MC). Please see https://doi.org/10.1007/s00226-018-1007-0 and https://doi.org/10.1007/s00226-016-0883-4 . These citations should be included and it should be noted that the DVS data may have extremely large errors.

The work of Simpson, which is also used in the Wood Handbook, has been shown to be based upon unsound scientific data. Please see https://doi.org/10.2737/FPL-GTR-229 . It may be important to note that in additon to overestimating the MC because it is for non-modified wood, it is also not a reputable source of data for scientific purposes.

Editorial- it appears the figures are no longer linked to the figure call-outs and "ERROR REFERENCE SOURCE NOT FOUND" is sprinkled throughout the document.

Suggestion- the data for each measurement location in tabular format is hard to read and is too much information for the paper. These could be included in supplementary information with the key findings included in the paper graphically.

Reviewer 2 Report

  1. The study is original and it does reflect substantial experience with the subject. The paper is clear and reads well. The paper deals with an interesting topic, but interest to the readers may be weak.
  2. The title is clear and reflects the study content. Some keywords are just a repeat of the title and I suggest that they should be changed.
  3. Materials, methods, and results are presented understandably. However, the conclusions are too weak. Moisture content was measured in different locations on the façade (elements of construction, distance above the ground, etc.), usually in two points. Therefore, conditions (for example wind-driven rain or water traps) for thermally modified and untreated wood were different. Authors write about differences between MC of thermally modified and untreated wood. Line 285: “…considerably lower”. The results can be compared? The results are significant statistically?
  4. What is the connection between decay (fungi) and the aim of the article? This is not clear. For example lines, 135-143. The shortening of the text is recommended.
  5. Figures are of low quality.
  6. An error occurred in many places in the text. For example lines: 71, 193, 196, 200. Before sending, check the text.
  7. Each wood relative air humidity responds to a determined MC equilibrium. From the viewpoint of physics, the equilibrium of wood's MC is a state in which the pressure of vapor in wood is equal to the pressure of vapor in the air. Once the equilibrium has been achieved the exchange of moisture between the material and the environment ceases. Hence the relative humidity of air is a key factor for wood moisture content and dimensionally stable. The key benefit of thermally modifying wood is to make abundant, non-durable wood species, more durable and dimensionally stable, etc. (see introduction). The obtained results are not novelty. The idea of this study was to assess the performance of a façade made of thermally modified wood with long time moisture monitoring. I suggest rebuilding text, rethinking some parts of the text and send again for review.